# NANOBODIES^®^: A Review of Generation, Diagnostics and Therapeutics

**DOI:** 10.3390/ijms24065994

**Published:** 2023-03-22

**Authors:** Bo-kyung Jin, Steven Odongo, Magdalena Radwanska, Stefan Magez

**Affiliations:** 1Laboratory for Biomedical Research, Ghent University Global Campus, Incheon 21985, Republic of Korea; bokyung.jin@ugent.be (B.-k.J.); steven.odongo@ghent.ac.kr (S.O.); magdalena.radwanska@ghent.ac.kr (M.R.); 2Department of Biotechnical and Diagnostic Sciences, College of Veterinary Medicine, Animal Resources and Biosecurity, Makerere University, Kampala 7062, Uganda; 3Center for Biosecurity and Global Health, College of Veterinary Medicine, Animal Resources and Biosecurity, Makerere University, Kampala 7062, Uganda; 4Department of Biomedical Molecular Biology, Ghent University, B-9052 Ghent, Belgium; 5Laboratory of Cellular and Molecular Immunology, Vrije Universiteit Brussel, B-1050 Brussels, Belgium; 6Department of Biochemistry and Microbiology, Ghent University, B-9000 Ghent, Belgium

**Keywords:** NANOBODY^®^, NANOBODY^®^ generation, NANOBODY^®^ production, diagnostics, therapeutics

## Abstract

NANOBODY^®^ (a registered trademark of Ablynx N.V) molecules (Nbs), also referred to as single domain-based VHHs, are antibody fragments derived from heavy-chain only IgG antibodies found in the *Camelidae* family. Due to their small size, simple structure, high antigen binding affinity, and remarkable stability in extreme conditions, nanobodies possess the potential to overcome several of the limitations of conventional monoclonal antibodies. For many years, nanobodies have been of great interest in a wide variety of research fields, particularly in the diagnosis and treatment of diseases. This culminated in the approval of the world’s first nanobody based drug (Caplacizumab) in 2018 with others following soon thereafter. This review will provide an overview, with examples, of (i) the structure and advantages of nanobodies compared to conventional monoclonal antibodies, (ii) methods used to generate and produce antigen-specific nanobodies, (iii) applications for diagnostics, and (iv) ongoing clinical trials for nanobody therapeutics as well as promising candidates for clinical development.

## 1. Introduction

Conventional antibodies consist of two heavy chains and two light chains adding up to a total molecular mass of 150 kDa in the case of IgGs. Each heavy chain consists of three constant domains (CH1, CH2, and CH3) and a variable domain (VH), while each light chain consists of a constant domain (CL) and a variable domain (VL). In 1989, Professor Raymond Hamers-Casterman of the Vrije Universiteit Brussel (VUB) made a serendipitous discovery, which was the existence of antibodies derived from *Trypanosoma evansi-*infected dromedary camels that lacked light chains. These light chain-devoid antibodies, referred to as heavy-chain only antibodies (HCAb), are found in animals of the *Camelidae* family and are characterized by the absence of the CH1 domains, needed for light chain pairing. Hence, they only had two heavy chains, each possessing a single variable antigen-binding (VHH) domain [1]. Remarkably, despite their truncated nature, these HCAbs displayed the ability to bind to a wide range of antigens, as shown through radioimmunoprecipitation and western blotting [1].

Subsequent successes in determining the crystal structure of the VHH domain of the HCAbs, as well as proving the feasibility of selecting, identifying and recombinantly expressing the VHH domain alone (hereby referred to as a single domain antibody or NANOBODY^®^ (Nb) which is a registered trademark of Ablynx N.V) have spurred on this field beyond the scope of trypanosome infections [2,3,4]. A little over three decades on, Nbs are now not only widely used in many research fields but they are also the subject of interest to companies for their various diagnostic and therapeutical applications.

## 2. Unique Structural Features of Nanobodies

Camelid HCAbs consist of a fragment crystallizable (Fc) region, homologous to that of conventional antibodies, directly joined to a fragment antigen-binding (Fab) region consisting of a single VHH domain [1,5,6]. The absence of both the CH1 domains and light chains results in a reduced molecular mass of 90 kDa as compared to the 150 kDa of conventional IgGs [7]. The VHH fragment of a HCAb has dimensions roughly of 2.5 × 4.0 nm and a molecular weight of 15 kDa, and can be cloned and recombinantly expressed as a monomeric Nb that is capable of binding to a wide antigen repertoire [8,9,10]. Nbs and VH domains of conventional antibodies share some similar structural traits. Both antibody fragments consist of four conserved framework regions (FR), and three hypervariable complementarity-determining regions (CDR) responsible for determining antigen specificity [7,11,12]. These seven regions (FR and CDR) fold into two β-sheets, one consisting of four β-strands and the other of fice β-strands, with the CDRs located in between the β-strands and gathered at the N-terminal of the Nb to form the antigen-binding site, also referred to as the paratope [7,11,13]. However, there also exists multiple structural dissimilarities between Nbs and conventional VH fragments, that provide a range of advantages to Nbs when compared to other monoclonal antibodies (mAb) or mAb fragments such as single-chain variable fragments (scFv).

While Nbs demonstrate antigen-binding capacity as a monomeric entity with only three CDRs, the VH domain of a conventional mAb is typically conjoined to the VL domain, thereby requiring a total of six CDRs to exhibit full antigen-binding capability [12]. Although it is expected that there is a limitation to the range of antigens that Nbs are capable of binding to, because of a smaller paratope due to the absence of the VL domain, this is circumvented by an enlarged CDR1 which serves to not only increase paratope size but also has been found to result in a wider range of loop architectures that do not conform to the canonical structures found in those of other VH domains [14,15]. Furthermore, Nbs also exhibit an extended CDR3 loop that greatly contributes to the diversity of antigens that the Nbs are able to bind to [16,17]. It has been shown that the CDR1, CDR2 and CDR3 of mAb VH and VL domains contribute equally to the binding of an antigen, whereas the CDR3 of Nbs have significantly higher contact proclivity with the antigen and act as the main contributor to binding site affinity [18,19]. Sequence and structure analyses have found that the diverse antigen-binding repertoire of Nbs does not stem from increased sequence variation of the CDR1 and CDR2, but rather the structural variation of the CDR1 and increased length as well as variation of the CDR3, thereby making up for the loss of the VL domain [19,20]. These two extended hypervariable regions (CDR1 and CDR3) of Nbs are often interconnected by a disulfide bond [21].

Another two key structural characteristics of Nbs are the amino acid residue composition of FR2 and the general convex shape of the paratope. While the FR2 of conventional mAb VH domains are largely made up of conserved hydrophobic amino acid residues at conserved positions, namely V37/G44/L45/W47, due to the necessary interaction with the VL domain for functionality, the FR2 of Nbs typically consist of hydrophilic amino acid residues, namely F37/E44/R45/G47, that provides an explanation for their ability to exist alone as a soluble monomer [11,21,22,23]. It should also be noted that an amino acid residue substitution of L11 into S11 in the FR1 region is commonly observed (more commonly observed in dromedary VHH sequences than in llama VHH sequences) [17]. Furthermore, unlike the concave or flat paratopes that are commonly exhibited by the VH–VL domains of mAbs, the paratopes of Nbs exhibit a convex paratope largely contributed by their extended CDR3 loop [11,24,25]. A diagrammatic representation and list highlighting the main structural differences between conventional m(IgG)Abs, HCAbs, and Nbs can be found in Figure 1A and Figure 1B, respectively.

## 3. Desirable Properties of Nanobodies over Conventional Monoclonal Antibodies

The unique structure of Nbs offer multiple advantages over conventional mAbs. Their small size, convex shape, and extended CDR3 confer onto them paratopes that can bind to concave sections of antigens that are often considered obstructed, and thereby in-accessible to the larger conventional mAbs [26,27]. Contrary to expectations, the Nb minute size does not hamper binding affinity as these antibody fragments are capable of exhibiting equilibrium dissociation constants within the nano- and picomolar range, putting them on par with other mAbs [7,28]. Furthermore, the smaller size of Nbs results in improved tissue penetration, allowing for the creation of Nb-drug/tracer conjugates that can penetrate tumors which conventional antibody-drug conjugates are unable to. Hence, this allows for more specific and efficient drug delivery, as well as improved imaging capability [29,30]. Additionally, Nbs have been shown to be able to cross the blood–brain barrier, both by natural means and artificial ones such as physical and chemical methods. They also do so in conditions of a compromised blood–brain barrier due to neurological pathologies [31]. Various studies have been carried out to provide proof of concepts that Nbs and Nb-conjugates can be put to imaging and therapeutic applications against a whole range of pathologies that primarily affect the central nervous system. This has been the case for glioblastoma, the meningoencephalitic stage of trypanosomosis, and auto-immune encephalomyelitis [31,32,33,34,35,36,37]. This key feature provides a potential solution to the persistent problem of the strict regulation that mAbs face when attempting to cross the blood–brain barrier.

In addition to the advantages already conferred upon them due to their smaller size, Nbs possess a whole host of favorable biochemical characteristics. Firstly, Nbs exhibit remarkable stability when exposed to high temperatures for prolonged periods which is in part due to their extended CDR3 and their ability to refold after denaturation [4,38,39,40,41]. It has also been shown that thermal stability can be further increased by means of selecting Nbs with an additional second disulfide bond, extending the length of the CDR3, and mutations of specific amino acid residues at the N-terminal [42,43]. Secondly, not only are Nbs highly soluble due to their hydrophilic FR2, thereby preventing aggregation and al-lowing for their function as a monomer, but they are also able to remain stable in the presence of proteases and display resistance to pH changes [23,39,44,45]. Combined, these traits present a possibility that Nbs can be administered through alternative routes such as oral or intraperitoneal delivery [26]. Interestingly, multiple studies have shown that inoculation with various Nbs raises no immune response, indicating that Nbs are either nonimmunogenic or possess a low immunogenicity [7,46,47,48,49]. This makes Nbs ideal potential candidates for drug development as they have a low risk of triggering an adverse side reaction during clinical trials. Additionally, it is possible to further mitigate the risk of adverse side reactions by methods such as mutating camelid specific amino acid sequences of Nbs to that of their human VH counterpart (i.e., humanization) or selecting Nbs from synthetic Nb libraries that have a pre-humanized scaffold with diversity achieved via CDR randomization [50,51].

Lastly, an additional advantage is that while conventional mAbs are (i) complex in structure, (ii) commonly post-translationally modified, (iii) require complex eukaryotic expression systems, and (iv) have complex purification steps, Nbs are comparably simpler to modify, produce, and purify [8]. By virtue of their smaller size and monomeric structure, Nbs are ideal candidates for multimerization which allows for multivalency, multiparatopicity, and multispecificity, which can increase avidity and enable binding of multiple antigens, respectively [8,52,53,54,55]. Nbs can also be conjugated to a variety of molecules such as fluorescent proteins or Fc regions for various applications such as imaging or effector function generation [56,57]. Furthermore, Nbs can also be produced by conventional prokaryotic expression systems such as *Escherichia coli* (*E. coli*) as well as eukaryotic systems such as *Saccharomyces cerevisae* and *Pichia pastoris* [58,59,60]. Expression in plant and mammalian systems have also been reported [61,62]. Finally, Nbs can also be easily purified through immobilized metal affinity chromatography (IMAC) when conjugated to a hexahistidine tag (his_6_ tag), or—based on their homology to human VH domains—through protein A affinity chromatography [63,64]. These combined advantages allow for the comparably economical production of Nbs for diagnostics/therapeutics with a possibly greater range of antigen recognition compared to their conventional mAb counterparts. However, it should be noted that the performance of Nb recognition of linear epitopes is usually inferior to that of mAbs. A condensed list of advantages Nbs possess over conventional mAbs can be found in Figure 1C.

Although Nbs possess many desirable properties, it would be remiss to not state their limitations. The first disadvantage of Nbs is that their monovalent format often requires modification, i.e., multivalency, for them to display sufficient therapeutic as well as diagnostic capacity, thereby often adding an additional complex step. The second disadvantage is that their small size of 15 kDa results in low serum persistence or rapid renal clearance as the threshold for glomerular filtration is 50–60 kDa, thus presenting a disadvantage in diagnostic screening and therapeutic applications [24]. A possible way to circumvent this is the conjugation of the Nb to polyethylene glycol or albumin [8]. Thirdly, Nbs lack an Fc region and therefore cannot exert effector functions that are associated with this moiety. However, this issue can be resolved by conjugating to a Fc region to increase the therapeutic capacity [57]. Additionally, the current most conventional way of generating Nbs requires the use of camelids, associated with far more housing space than the production of mAbs which are commonly generated by immunizing mice. The required use of camelids also presents the drawback of requiring humanization strategies for therapeutic Nbs to minimize the risk of adverse immunogenic side reactions. This, however, comes with the risk of losing their desirable refolding ability advantage or sacrificing antigen binding affinity [50,51]. Lastly, although Nbs excel at recognizing concave/hidden/conformational epitopes due to their extended CDR3, this may come at the cost of being far less proficient at recognizing flat/linear or convex epitopes thereby reducing the epitope range Nbs are able to recognize. Although such a method has not been validated, it may be possible to initially denature conformational proteins and select for binders so that Nbs may be generated against such epitopes.

## 4. Nanobody Generation and Production

### 4.1. Generation

Antigen-specific nanobodies can be selected from three different sources, namely immune, naïve, and synthetic libraries [8,65]. Immune libraries are typically obtained by immunizing a domesticated animal of the *Camelidae* family (such as alpacas or dromedaries) with a target antigen, a minimum of four times, generally over the course of two months [65]. It is also possible to use transgenic mice that have been transformed to produce HCAbs in situations where the target antigen is limited [66,67,68]. Lymphocytes are then purified from extracted blood for mRNA extraction, before cDNA conversion [8,65]. A two-step polymerase chain reaction (PCR) approach is then applied by first amplifying from the leader sequence to a conserved region within the CH2 domain of all IgGs, followed by agarose gel electrophoresis to select for sequences from HCAbs (smaller amplicons compared to those of regular antibodies are obtained due to the absence of the CH1 domain), and finally amplifying the VHH sequence by primers targeting viable restriction enzyme sites [65]. Lastly, the obtained amplicons are ligated into a vector and transformed into an appropriate expression system, typically *E. coli* although it is possible to use other expression systems such as yeast or mammalian systems [58,60,62,65]. A typical ideal immune library should contain at the minimum 10^7^ individual unique transformants [65].

Once the Nb library is obtained, target antigen-specific Nbs must be selected and retrieved. The most common method employed is through the use of phage display [8,63,65]. The Nb library cells are infected with M13 phages, which results in the production of phages that contain the Nb DNA sequence. In addition, these phages will display the Nb fused to their coat proteins, which are then introduced to a microtiter plate with the affixed target antigen. Positive binders are eluted via a pH shock (through the addition of triethylamine at pH 10). Multiple rounds of biopanning allow for the discarding of phages displaying weak affinities, hence eliminating non-binding Nbs [65,69]. Next, the phages are once again transfected into new *E. coli* cells, where they are cultivated in a medium, using ampicillin as a selection marker. This allows for mass culturing of the positive binders, so they can be validated via the use of an enzyme-linked immunosorbent assay (ELISA), before being sequenced to determine the Nb nucleotide sequence [69,70]. Finally, to determine the highest affinity binders and exclude the weaker binders in a pool of candidates, without the Nb expression level skewing results, surface plasmon resonance (SPR) is employed [65]. Here, the target antigen is affixed onto the dextran layer of a SPR chip and the candidate Nb is passed over the chip, allowing for the measurement of the association rate (kon) as it binds and disassociation rate (koff) as it is washed off [65]. In general, kon/koff rates of 10^5^–10^6^ and 10^−2^–10^−4^ respectively as well as equilibrium dissociation constants in the nano-/picomolar range are ideal [7].

For the long-term storage of bacteria stocks that produce Nbs that exhibit satisfactory target antigen binding capability, glycerol stock of the Nb expressing bacteria is often made. Phagemids or plasmids containing the Nb nucleotide sequence from the cells expressing the desired Nb are extracted and commonly transformed into *E. coli* WK6 cells in media, which then have an equal volume of glycerol, glucose and ampicillin added and thus can then be stored for extended periods for future production [69,71]. An overview scheme of the conventional method of generating antigen specific Nbs is provided in Figure 2. While the above method has been optimized at the VUB, alternative methods that use different technologies for the production of

Nbs can be envisaged and have been described by others [58,59,60,61,62,72,73,74,75,76].

### 4.2. Modifications to the Standard Nanobody Generation Protocol

Although the above Nb generation protocol is well-optimized and widely used, various modifications can be employed in the case of limitations, specific requirements, or unique situations. In the event where immunization is undesirable, naïve or synthetic libraries can be employed. Here, the HCAbs are extracted from an unimmunized animal or an ideal VHH scaffold is selected, and variation is introduced to the CDRs via amino acid randomization. However, this comes with the drawback of requiring a larger library to the scale of >10^9^ unique transformants and, in the case of naïve libraries, a large amount of blood is needed for the required number of lymphocytes [63,65]. It should be noted that synthetic libraries also have the advantage of being able to generate Nbs against antigens that are either non-immunogenic or toxic. The technology also eliminates the ethical concerns that come with the camelid vaccination protocol [70].

As an alternative to the method described above, it is also possible to use several other selection systems such as ribosome display, bacteria display, and yeast display. This circumvents the drawback of M13 phages “sticking” to the microtiter plate and resulting in a reduction of efficiency when attempting to select a high affinity binder [72,73,74,75,76]. Different selection systems also possess unique advantages, such as the selection of positive binders using fluorescence-activated cell sorting (FACS) in the case of yeast and bacteria display. Although this requires a FACS infrastructure and introduces additional technicalities, it allows for the normalization of Nb expression on the surface to binding capability, thereby allowing for the identification of Nbs with high affinity and expression [65,77]. Furthermore, ribosome display has the unique advantage of not requiring the transformation of microbial cells, thereby allowing for the rapid construction of a library via multiple rounds of PCR amplification [74]. However, it should be said that each method also presents unique drawbacks. These can include the presence of interfering factors such as nucleases for ribosome display, slower growth rates for bacteria display, and lower transformation efficiency as well as longer required culturing times for yeast display [74,75].

### 4.3. Production and Purification

The standard production of Nbs conventionally employs the use of the *E. coli* expression system although, as stated before, other prokaryotic and eukaryotic systems may be used depending on specific needs of the Nb and its application. A luria broth (LB) medium (supplemented with 2% (wt/vol) glucose, 100 μg/mL ampicillin, and 1 mM magnesium chloride), that has been inoculated with glycerol stock of the Nb-expressing *E. coli* cells, is incubated overnight before being used in turn to inoculate a terrific broth (TB) medium (supplemented with 0.1% (wt/vol) glucose, 100 μg/mL ampicillin, and 1 mM magnesium chloride) [71,78]. After reaching sufficient growth (3 to 4 h of incubation and measured by optical density of 0.6–0.8), Nb expression is induced by the addition of 1 mM (final volume) isopropyl beta-D-1-thiogalactopyranoside (IPTG) and followed by further overnight incubation [71,78]. Cells (pellets) are then collected via centrifugation (multiple rounds of 20 min at 4000 RPM), the periplasm is lysed by osmosis via the addition of Tris EDTA Sucrose (TES) buffer [consisting of 171.15 g sucrose, MW = 342.2; 5 mL Na2EDTA 100 mM; and 200 mL Tris-HCl 1 M (pH 8.0) per liter] and followed by TES/4 buffer (1 part TES buffer and 3 parts distilled water) [71,78]. Thereafter, the cell lysate is centrifuged for 60 min at 4000 RPM to collect the supernatant (periplasmic extract) [69,71,78]. Should a yeast expression system or vector modified to produce secretory Nbs (such as one utilizing the hemolysin secretion system) be used instead of the standard *E. coli* system, purification is simplified as the Nb is secreted [79,80,81].

The purification of Nbs is often carried out using IMAC targeting a his_6_ tag linked to the Nb primary sequence. The periplasmic fraction is passed over an IMAC column loaded with a solution of nickel resin, washed with phosphate-buffered saline (PBS) solution (either pure or containing a low concentration (0.02 M) of imidazole to elute weakly binding proteins), and finally the Nb is eluted through the addition of (0.5 M) imidazole solution [71,78]. A further step of size-exclusion chromatography can be applied to obtain a greater purity level [71,78]. In some cases, other chromatographic methods of purification can be applied such as protein A affinity chromatography or cation exchange chromatography [64,82]. Verification and quality control of the obtained Nb protein is then performed through SDS-PAGE, western blotting, and differential scanning fluorimetry among others [78]. An overview of a scheme for the production and purification of a Nb starting from its glycerol stock is provided in Figure 3. The production protocol discussed above which uses the *E. coli* expression system is considered to be standard. This expression system has the advantage of being highly customizable in its parameters (incubation temperature, culture times, reagent concentrations, and media used) and a change to more optimal conditions can result in the improved periplasmic expression and reduced loss of antibiotic pressure for the production of intractable Nbs [78]. Thus, it is possible to circumvent the required usage of other complex expression systems and improve the production yields of intractable Nbs while maintaining the use of the simpler *E. coli* expression system [78].

Note that where different expression systems are used instead of the conventional *E. coli* one, the parameters, reagents and methods employed in the production, extraction, and purification of the Nb will also differ.

## 5. Diagnostic Applications of Nanobodies

Due to their unique characteristics and advantages over conventional mAbs, there is much ongoing research and development into the application of Nbs in the broad field of diagnostics. Figure 4 shows a general overview of the different formats in which Nbs can be applied, the types of diagnostic methods they can be used for, and the applications for which they are being considered.

### 5.1. Nanobody Immunoassay Format: Lateral Flow Immunoassays and Diagnostic ELISAs

The diagnosis of disease via microscopy-based and molecular-based methods, such as PCR or loop-mediated isothermal amplification, offers a high degree of sensitivity (the ability to correctly determine true positives) and specificity (the ability to avoid false positive readouts) [83]. Unfortunately, such methods require specialized equipment (thereby possibly rendering the approach unaffordable in a resource-poor setting), specialized training, and can only be carried out at specific locations—thus resulting in failure to fulfill the entire ASSURED criteria, whereby the ideal diagnostic test is affordable, sensitive, specific, user-friendly, rapid/robust, equipment-free, and deliverable to those who need it [83]. For this reason, the lateral flow immunoassay (LFIA) diagnostic test is ideal as it can satisfy the ASSURED criteria.

A lateral flow immunoassay is a rapid diagnostic test that specifically detects the presence of an antigen of interest within a mixture. It works via the introduction of the sample to the sample pad, and—via capillary action—it results in the sample mixture flowing through a conjugate release pad (containing antigen-specific antibodies conjugated to colored or fluorescent markers), test line (containing antibodies capable of binding and holding the conjugate-complexed antigen), and the control line (to ensure that the LFIA is functional) before reaching the absorbent pad at the end [84]. Should the antigen of interest be present, a ‘sandwich’ is formed at the test line during sample flow, which can be visibly observable within five minutes. Nbs present an ideal replacement for the antibodies used in a LFIA as they (i) are highly stable and heat resistant thereby allowing for longer storage (a great advantage for distribution in developing countries), (ii) possess a greater paratope repertoire thereby allowing for binding to epitopes that are inaccessible to conventional antibodies as well as preventing competition with host antibodies, (iii) lack a Fc region, which prevents any possible cross-reactions with host antibodies within the sample, (iv) can be easily modified into a multivalent construct, and (v) be adsorbed onto gold nanoparticles, which allows for more detection options in a LFIA setup [18,41,85,86,87].

Much work has been done by the Cellular and Molecular Immunology (CMIM) research unit of the VUB into the development of a LFIAs. Published work includes the detection of the *Trypanosoma congolense* glycolytic enzyme pyruvate kinase in plasma, to diagnose active trypanosomosis infections [85]. This was achieved via generating a Nb library against the *T. congolense* secretome, isolating a pair of Nbs against pyruvate kinase, constructing a sandwich ELISA using said Nb pair, and translating the ELISA into a LFIA format [85]. The experimental LFIA displayed a sensitivity of 80% and a specificity of 92%, indicating that further development could possibly result in the production of a commercial LFIA [85]. Additional work applying similar approaches has resulted in identifying other Nbs against glycolytic enzymes such as enolase and aldolase for the diagnosis of *T. evansi* and *T. congolense*, respectively [88,89,90]. A LFIA for the detection of human norovirus infection has also been reported with a promising sensitivity of 80% and specificity of 86% against four different norovirus antigens representative of four different strains [91].

Diagnosis of infection is not the only application that LFIAs are useful for. Nbs can also be implemented in the detection of molecules such as recombinant human interferon a2b, a therapeutical molecule used in the treatment of hepatitis B/C. In this case, LFIA development targeted a way to implement a rapid quality control tool for the detection of faulty or counterfeit pharmaceuticals [92]. Another application is the use of LFIAs for the rapid detection of pathogens and toxins in food and the environment. This has the potential to reduce cumbersome transport of samples to a laboratory by providing an easily interpretable result in the field. As proof of concept, LFIAs were developed using Nbs targeting aflatoxin B1 in almond milk, and 3-phenoxybenzoic acid in milk and lake water [93,94].

ELISA setups can also be used as laboratory based diagnostic methods without the need to translate to a LFIA format. Such a setup often comes with the advantage of a greater limit of detection as well as the ability to optically quantify said target antigen. The obvious disadvantage is that such a diagnostic test does not meet the ASSURED criteria as specialized training (to perform the ELISA), necessary equipment (specific reagents and a plate reader), and time are necessary to perform the test. Nevertheless, a number of Nb-based sandwich ELISAs targeting (i) biomarkers for cancer diagnosis (glypican-3 and fibrinogen-like protein 1), (ii) toxins within the environment and food (tetrabromobisphenol A and ochratoxin A), and (iii) bacteria for food safety analysis (*Listeria monocytogenes*, *Salmonella enteritidis*, *Staphylococcus aureus*) have been developed, indicating the potential of Nbs in such ELISA diagnostic tests against a wide variety of antigens [95,96,97,98,99,100,101]. A Nb-based competitive ELISA for the diagnosis of foot and mouth disease antibodies within cattle has also been reported [102].

### 5.2. Biosensors

A relatively new application where Nbs have been implemented successfully is the field of biosensor development. Conventionally, a biosensor is a device with a bioreceptor specific to a target antigen is affixed to a semiconductor. In the event of a binding between the receptor and the target antigen, an electric potential change is recognized and amplified into a measurable signal [103]. The greatest advantage of a biosensor is that results are instantaneous, in the scale of seconds, thereby outperforming the ELISA diagnostic test and even LFIAs. Nbs are an ideal candidate for bioreceptors, as their ability to withstand high temperatures as well as pH changes allows for storage in less-than-ideal environments [104]. A Nb utilizing biosensor for the detection of fibrinogen, a biomarker for cardiovascular diseases, was proven to be viable and shown to have a satisfactory limit of detection at 0.044 μg/mL, with its function confirmed using an international standard of fibrinogen plasma [105]. Nbs have also been shown to be candidates for the development of biosensors for the use of detecting severe acute respiratory syndrome coronavirus-2 (SARS-Cov-2) and Middle East respiratory syndrome coronavirus spike proteins, making them a promising tool in the fight against the COVID-19 pandemic [106,107].

### 5.3. In Vivo Diagnostic Imaging

In vitro diagnostics, such as the types that are discussed above, are vital in the field of disease diagnostics. However, they still possess limitations that render them unsuitable for the detection of certain diseases. This is the case for cancer, which can be localized to a certain part of the body and hence can be undetectable in bodily fluids. In vivo diagnostic methods, such as medical optical imaging and nuclear imaging techniques, positron emission tomography (PET) and single photon emission computed tomography (SPECT), make use of fluorescently or radioactively labelled antibodies to visualize aberrations within the body, namely tumors. Such techniques are more suitable as they are not only capable of diagnosis alone but also of monitoring the progression of said aberration. The small size of Nbs allowing for rapid tissue penetration, ease of modification/conjugation, and rapid renal clearance therefore make Nbs an ideal candidate in the development of fluorescent or radioactive target specific tracers for in vivo diagnostic methods [8,30,56,57].

Although optical imaging methods do not penetrate as deep as nuclear imaging methods, due to the use of fluorescently labelled tracers instead of radioactively labelled ones, optical imaging is considered safer as nonionizing radiation is utilized. The technology also has the added advantages of being cheaper as well as being able to produce real-time information due to their faster result times [108]. An anti-epidermal growth factor receptor (EGFR) (a biomarker for human epithelial cancers) Nb, referred to as 7D12, has been conjugated to fluorophore IRDye800CW and compared to the commercial mAb treatment Cetuximab [109]. Results showed that 7D12 exhibited not only a faster measurable signal when compared to Cetuximab at 30 min post-injection, but also exhibited a greater penetrative capability into tumors indicating its promise as a fluorescent tracer for optical imaging [109]. Furthermore, a dual tracer approach was tested using an anti-human growth factor receptor 2 (HER2) Nb in tandem with an anti-carbonic anhydrase IX Nb, which showed a greater target-to-background ratio than when only either one was used, which points to a better early diagnosis capability as well as tumor characterization ability [110]. Moreover, a Nb against V-set and immunoglobulin domain-containing 4 conjugated to fluorescent dye Cyanine 7 has been tested for its ability to monitor arthritis in vivo, and was able to show high target-to-background signals 3 h post-injection, indicating its ideal use for optical imaging [111].

Regarding PET/SPECT nuclear imaging, both methods have the advantages of being noninvasive and quantitative, while possessing greater penetration capability and sensitivity when compared to optical imaging [108]. Hence, there has been much work carried out to prove their application as radioactive tracers used in both methods. As proof of the viability of using Nbs as radioactive tracers in PET/SPECT, there have been many studies demonstrating the use of radioactively labelled Nbs against HER2, a common breast cancer biomarker. In one study, a Ga-68 conjugated Nb was preclinically tested and found to have high tumor-to-muscle/blood ratios 1 h post injection, thereby rapidly generating high contrast PET imaging results [112]. This promising diagnostic candidate Ga-68 Nb has completed a phase 1 clinical trial and is now undergoing a phase 2 trial that is expected to finish in 2023 [113,114]. Other anti-HER2 Nbs have also been labelled with various radioactive labels such as F-18 and Lu-177, with the former demonstrating high tumor-to-muscle/blood ratios allowing for the generation of high contrast PET images, and the latter demonstrating lesser radiation burden to non-target organs compared to a conventional mAb Lu-177 conjugate [115,116]. Radioactively labelled Nbs have been developed for the detection of not only other cancer biomarkers, such as EGFR and fibronectin, but also different diseases such as atherosclerosis—indicating that Nbs have a promising future in the field of in vivo diagnostic methods [117,118,119,120].

## 6. Therapeutical Applications of Nanobodies

Nb-based therapeutics have received much interest in recent years, with many undergoing various phases of clinical trials. Three therapeutic Nbs have already been approved for clinical use. Table 1 gives a non-exhaustive overview of the Nb-based therapeutics discussed in this review.

### 6.1. Nanobodies Against Cancer

The excellent tumor penetration capabilities, ability to recognize unique antigens, and other advantages of Nbs make them a promising candidate in the field of cancer therapeutics. One avenue of Nb cancer therapy would be the development of chimeric antigen receptor (CAR) T-cells expressing Nbs specific to tumor antigens. T-cells are extracted from a patient and genetically modified to express a tumor antigen specific Nb before being reinfused into the patient, thus allowing the T-cell to bind and neutralize tumor cells through mechanisms such as the release of cytotoxic molecules, induction of apoptosis through tumor necrosis factor receptor recognition, and secretion of inflammatory cytokines [121]. In 2018, the results of phase 1 clinical trial for a Nb CAR T-cell candidate that targeted the cancer biomarker B-cell maturation antigen for the treatment of refractory/relapsed multiple myeloma, showed promising results [122]. It completed phase 1 clinical trials successfully and underwent further phase 2 trials [123,124]. A follow-up study done 4 years after initial administration also showed favorable long-term safety and durability of the modified T-cells [123]. Based on the success of the phase 1 and 2 clinical trials, this Nb CAR-T cell candidate, rebranded as Ciltacabtagene autoleucel, was approved for the treatment of multiple myeloma by the FDA in February 2022 [125]. Many other proof-of-concept studies have also been carried out for developing Nb expressing CAR T-cells that target other cancer biomarkers such as CD20, EGFR, and HER2 [126,127,128].

Another avenue of cancer therapy is the use of multivalent Nbs targeting specific receptors on tumor cells. The aim here is to directly trigger programmed cell death through the activation of the caspase-3 and caspase-8 [129]. A study by Huet et al. showed that a pentavalent Nb agonist to death receptor 5 was able to induce greater levels of tumor cell apoptosis than a tetravalent Nb in vitro, and exhibited a greater anti-tumor effect than a death receptor 5 mAb in vivo [129]. It is also possible to use Nbs targeting immune checkpoints (since blocking CD279 and CD274 un-suppresses T-cell apoptosis and upregulates antitumor responses) or to conjugate a tumor-antigen specific Nb to a toxin for endocytosis into the tumor cell (fusion of Pseudomonas exotoxin A to a Nb specific to vascular endothelial growth factor 2) [130,131]. Nbs can also be used in radionuclide therapy whereby a Nb against a tumor antigen is conjugated to a radiopharmaceutical for specific delivery to a tumor. An I-131 conjugated anti-HER2 Nb for the treatment of breast and gastric cancer has passed phase 1 and is now undergoing phase 2 clinical trials [132]. Lastly, a Nb-Fc conjugate against the chemokine receptor CXCR4 was shown to exert antibody-dependent cellular cytotoxicity and to complement dependent cytotoxicity effector functions on CXCR4 overexpressing tumor cells, with no side reactions to cells expressing none or low levels of CXCR4 [57].

### 6.2. Nanobodies Against Autoimmune Diseases

Perhaps the area where Nbs have found the greatest success regarding therapeutic application is the treatment of autoimmune disorders. In 2018, a landmark success for the field of Nb therapeutics occurred when Caplacizumab was approved by the European Union for the treatment of acquired thrombotic thrombocytopenic purpura, i.e., a rare blood-clotting disorder [133]. Shortly after, in 2019, Caplacizumab was also approved by the United States FDA for consumer prescription [134]. Another Nb therapeutic that has found its way onto the commercial market is Ozoralizumab, developed by Taisho Pharmaceuticals under license of Ablynx [135]. As of September 2022, Ozoralizumab, targeting tumor necrosis factor-alpha, is approved for the treatment of rheumatoid arthritis in Japan [135]. Vobarilizumab is another promising Nb drug for the treatment of rheumatoid arthritis, as well as systemic lupus erythematosus, targeting the interleukin-6 receptor. This Nb therapy is currently undergoing phase 2 clinical trials [136,137].

Other Nbs used to treat various autoimmune disorders are currently in various phases of clinical trials. These treatments include (i) Sonelokimab (M1095) targeting interleukin-17A/F for psoriasis, (ii) Gefurulimab (ALXN1720) targeting autoantibodies against acetylcholine receptors for generalized Myasthenia gravis, (iii) Nb M6495 targeting A Disintegrin and Metalloproteinase with Thrombospondin Motifs-5 for osteoarthritis, and (iv) Nb V565 against tumor necrosis factor for Crohn’s Disease [138,139,140,141]. To further illustrate the potential of Nbs in therapeutics against autoimmune disorders, potential Nb candidates against eosinophilic asthma and multiple sclerosis have also been identified [142,143].

**Table 1 ijms-24-05994-t001:** Table summarizing the therapeutic products that have been developed from nanobodies, showing the disease and/or condition they target, the antigen targeted, their clinical trial status, manufacturer, and references in the literature to the therapeutic Nbs discussed in this review.

Product Name	Disease and/or Condition Targeted	Target Antigen	Clinical Trial Status	Manufacturer	References
68-GaNOTA-Anti-HER2 VHH1 ^1^	Breast carcinoma	HER2	Phase 2	Universitair Ziekenhuis Brussel, Brussels, Belgium	[113,114]
Ciltacabtagene autoleucel, LCAR-B38M	Refractory/relapsed multiple myeloma	B-cell maturation antigen	Approved	Janssen Research & Development, LLC, Raritan, United States	[123,124,125]
131I-GMIB-Anti-HER2-VHH1	Breast carcinoma	HER2	Phase 1	Precirix, Brussels, Belgium	[132]
Calplacizumab	Acquired thrombotic thrombocytopenic purpura	Von Willebrand factor	Approved (USA, EU)	Ablynx, Ghent, Belgium	[133,134]
Ozoralizumab	Rheumatoid arthritis	Tumor necrosis factor-alpha	Approved (Japan)	Taisho Pharmaceuticals, Tokyo, Japan	[135]
Vobarilizumab (ALX-0061)	Rheumatoid arthritis, systemic lupus erythematosus	Interleukin-6 receptor	Phase 2	Ablynx, Ghent, Belgium	[136,137]
Sonelokimab (M1095)	Psoriasis	Interleukin-17A/F	Phase 2	Bond Avillion 2 Development LP, London, England	[138]
Gefurulimab (ALXN1720)	Myasenthia Gravis	Autoantibodies against acetylcholine receptors	Phase 3	Alexion Pharmaceuticals, Boston, United States	[139]
M6495	Osteoarthritis	A Disintegrin and Metalloproteinase with Thrombospondin Motifs-5	Phase 1	Merck KGaA, Darmstadt, Germany	[140]
Nb V565	Crohn’s Disease	Tumor necrosis factor	Phase 2	VHsquared Ltd., Copenhagen, Denmark	[141]
ARP1, VHH batch 203027	Diarrhea	Rotavirus	Phase 2	International Centre for Diarrhoeal Disease Research, Dhaka, Bangladesh	[144]
ALX-0171	Lower respiratory tract infection	Respiratory syncytial virus	Phase 2	Ablynx, Ghent, Belgium	[145]
LMN-101	Campylobacteriosis	*Campylobacter jejuni*	Phase 2	Lumen Bioscience, Inc., Seattle, United States	[146]

^1^ Denotes a Nb that is undergoing trials as a possible radioactive tracer for use in PET/SPECT diagnostic imaging but is not in itself a therapeutic product.

### 6.3. Nanobodies Against Infectious Diseases

Contrary to the wide range of Nb candidates for oncology and autoimmune disorders, a cursory search through the National Library of Medicine’s clinical trial database yields next to no therapeutic candidates against infectious diseases caused by viruses, bacteria, and parasites. Nevertheless, there are a few reports of Nb therapeutics against infectious diseases such as VHH batch 203027 targeting Rotavirus for the treatment of diarrhea and ALX-0171 for treating respiratory syncytial virus lower respiratory tract infections [144,145]. Unfortunately, no updates on the former have been released since 2013 and the latter candidate failed to produce significant results in a phase 2b trial [144,145]. A more recent therapeutic Nb candidate, LMN-101, for the treatment of *Campylobacter jejuni* infections has been reported and is currently undergoing phase 2 trials [146]. The lack of therapeutical Nbs does not mean that Nbs are unsuitable for the development of therapeutics for the treatment of viruses, bacteria, or parasites, but rather indicates that this field is still very much in its infancy.

Regarding viruses, the recent global outbreak of the SARS-CoV-2 has shown how versatile Nbs can be in the fight against this pandemic. Multiple groups were able to report the development of Nbs specific to the spike protein of the receptor binding domain of the virus, which is the ideal candidate for the development of therapeutics [53,55]. Furthermore, Nbs were shown to be viable for the recognition of hidden spike protein epitopes of five different strains, highlighting their potential in the protracted fight against SARS-CoV-2 [147]. Other recent promising Nbs against viruses include a Nb–single chain variable fragment conjugate carrying siRNA against the human immunodeficiency virus, and a bivalent Nb construct that exhibited exceptional binding affinity to the influenza H7N9 virus [148,149]. Nbs also show promise for the development of therapeutics against bacteria and bacteria-produced toxins. As an example, three high affinity Nbs against the internalin B surface protein were shown to be able to neutralize *Listeria monocytogenes* in vitro [150]. In addition, two bivalent Nb constructs, one bivalent for *E. coli* Shiga toxin-2a B subunit and the other bispecific for *E. coli* Shiga toxin-2a B subunit and cell surface protein intimin, were postulated to be novel neutralizers for the said *E. coli* virulence factors [151,152]. It has also been shown that *Lactococcus lactis* bacteria—modified to display on their surface Nbs specific to the fimbrial adhesins of *E. coli*—were able to reduce fecal *E. coli* shedding when orally fed to piglets [153].

Regarding the use of Nbs against parasites, several studies are worthwhile mentioning here. Firstly, it was discovered how a chimeric construct between an anti-trypanosome Nb and a truncated version of the human trypanolytic protein APOL1 was able to exert a potent anti-parasite effect in vivo [46]. Secondly, it has been shown that the use of large doses of high affinity Nbs against trypanosome variant surface glycoprotein (VSG) can induce trypanocidal effects [154,155]. There have also been multiple efforts to create VSG specific Nbs conjugated to carriers (either constructed of poly(lactic-co-glycolic acid), or chitosan) and loaded with the trypanocidal drug pentamidine for targeted administration via endocytosis. This approach was able to reduce the required curative dose multiple-fold [156,157]. Lastly, two high affinity Nbs against Pfs230, a surface protein expressed during the sexual phase of the malaria-causing *Plasmodium falciparum*, were able to decrease exflagellation and disrupt the sexual reproduction stage of the male microgametes—thereby opening up the possibility of using Nbs in the development of a transmission-blocking malaria vaccine [158].

### 6.4. Nanobodies Against Toxins and Venoms

A final area where Nbs have shown promise is the treatment of snake/scorpion envenoming. Various formats of Nbs for the treatment of *Androctonus australis hector* (Aah) scorpion toxin have been reported. Among those reported include (i) a bispecific Nb for AahI’ toxin, which was also proven to be capable of being expressed by the *Pichia pastoris* expression system, (ii) a Nb–human Fc conjugate against AahI’ toxin, and (iii) a humanized Nb against AahII toxin, which exhibited no affinity drawback [159,160,161]. Regarding snake envenoming, a Nb–Fc conjugate against α-Cobratoxin of the *Naja kaouthia* snake as well as multiple Nbs against different components within *Bothrops atrox* snake venom have been reported [162,163].

## 7. Conclusions and Future Outlook

Since their serendipitous discovery slightly over three decades ago, the field of Nbs has grown exponentially. The multitude of advantages Nbs possess over conventional mAbs and conventional mAb fragments has spurred research forward, and opened many alternate methods and techniques for the generation and production of application/situation-dependent Nbs. Nbs have found much success in various formats of diagnostic tools such LFIAs, diagnostic ELISAs, biosensors, and in vivo diagnostic imaging, not only for disease detection but also detection of food-borne pathogens or environmental toxins. Furthermore, in the past few years, the multiple therapeutic approvals for Nb treatment of cancers and autoimmune diseases have driven commercial and industrial interest. Hence, this field is set to grow further in the near future. Additionally, the potential for Nb therapeutics against infectious diseases has also been shown to clearly exist but still requires much research and development. Hence, it is certain that Nbs will play an important role in the development of next generation diagnostic tools and therapeutics in the years to come.

## Figures and Tables

**Figure 1 ijms-24-05994-f001:**
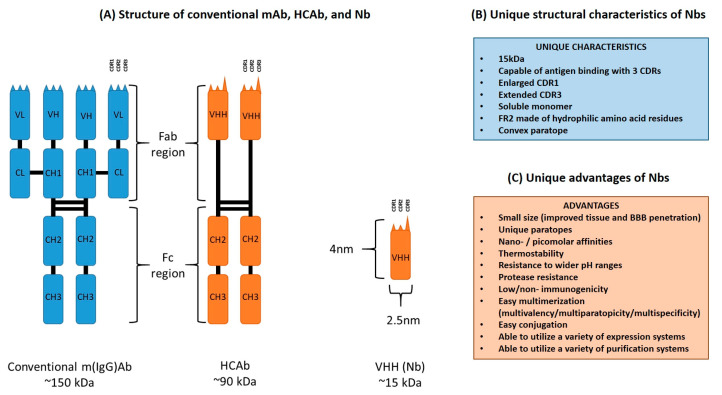
(**A**) Diagrammatic representation of the structure of conventional m(IgG)Ab, HCAb, and VHH (Nb), (**B**) an outline of the unique characteristics found only in Nbs, and (**C**) an outline of the advantages of Nbs compared to other mAb formats.

**Figure 2 ijms-24-05994-f002:**
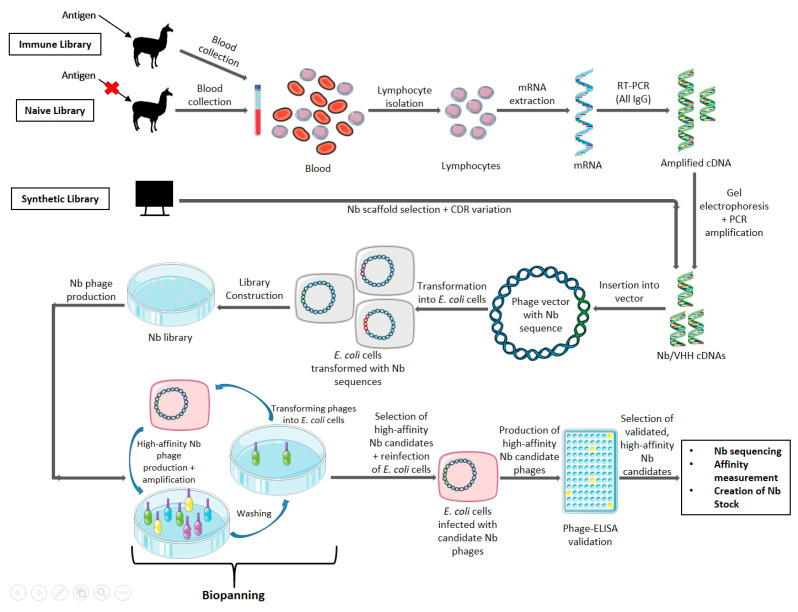
Schematic representation of the conventional method of generating antigen-specific Nbs using a phage display library. For an immune library, the camelid is inoculated with the antigen of interest, but no inoculation occurs for the generation of a naïve library. Blood is then collected, lymphocytes extracted, mRNA extracted, and a Nb sequence library is constructed via two rounds of PCR and a gel electrophoresis to select for the smaller Nb sequences. In the case of a synthetic library, a suitable Nb scaffold is selected, and the CDR are varied through amino acid randomization. The library is inserted into phage vectors and transformed into *E. coli* cells, for the production of phages containing the Nb nucleotide sequence and displaying the Nb on the outside. Multiple rounds of biopanning are then carried out to isolate the highest affinity binding phages, which are then used to reinfect *E. coli* cells for the creation of more phages with the high-affinity Nb. These phages are then validated using phage-ELISA; and Nb candidates can then be sequenced, have their affinity measured, and be used to create glycerol stock for future production [7,8,63,65,69,70,71].

**Figure 3 ijms-24-05994-f003:**
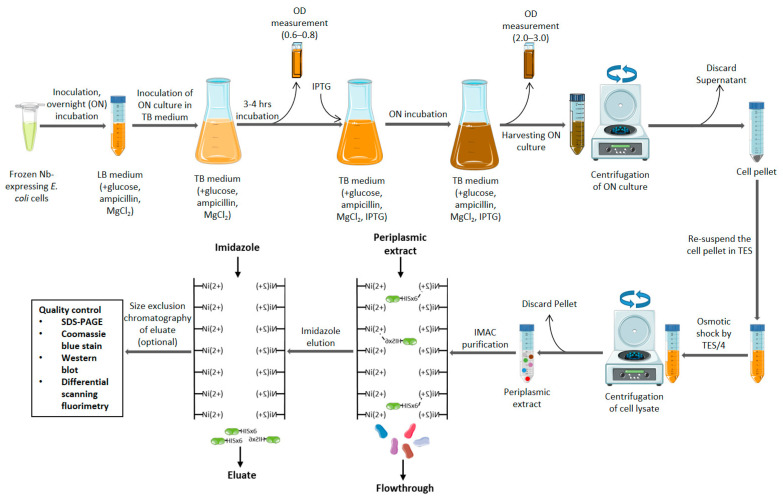
Schematic representation of the conventional protocol for the laboratory scale production of Nbs starting from glycerol stock (frozen Nb expressing *E. coli* cells). LB medium (supplemented with glucose, ampicillin, and MgCl_2_) is inoculated with the glycerol stock and incubated overnight. This LB medium is used to inoculate a TB medium (supplemented with glucose, ampicillin, and MgCl_2_) and incubated for 3–4 h. Nb expression is induced via the addition of IPTG and incubated overnight. After each incubation, the absorbance is measured to ensure that cell multiplication (absorbance limit 0.6–0.8) and Nb expression (absorbance limit 2.0–3.0) occurred satisfactorily. The cell pellet of the TB medium is then aggregated through multiple rounds of centrifugation at 4000 RPM (using a tabletop centrifuge with TB medium in 50 mL falcon tubes) for 20 min each round. The aggregated cell pellet then undergoes periplasm lysis through the addition of a recommended quantity of TES buffer followed by the addition of TES/4 buffer. The periplasmic extract (supernatant) is then harvested through a round of centrifugation at 4000 RPM for 60 min using a tabletop centrifuge with TB medium in 50 mL falcon tubes) and the pellet is discarded. The periplasmic extract is then purified through IMAC and the Nb is eluted through the addition of imidazole. It is also possible to add a further purification step by size exclusion chromatography. The Nb then undergoes quality control via SDS-PAGE, Coomassie blue staining, western blotting, and differential scanning fluorimetry [71,78].

**Figure 4 ijms-24-05994-f004:**
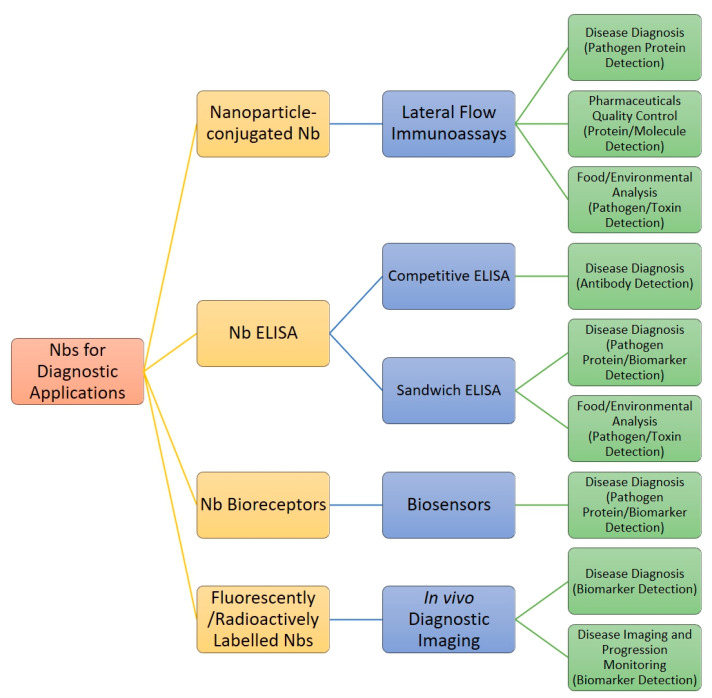
Flowchart representation of Nb use for diagnostics. Yellow represents the format in which Nbs are applied. Blue represents the method/device in which these Nb formats are used. Green represents the type of target (biomarker/protein/toxin/pathogen/molecule) for each of the assay formats.

## Data Availability

Not applicable.

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
