# Peer review of "NANOBODIES®: A Review of Diagnostic and Therapeutic Applications"

_ijms, 2023, doi:10.3390/ijms24065994_

Round 1
Reviewer 1 Report
The authgors have done an excellent job in summarizing a complex field. Regarding therapeutic applications, inclusion of information about Fc-mediated function of HCAbs in human/animal systems, attempts at Fc engineering, and serum persistence/PK (of both HCAbs and nanobodies), would greatly improve that section of the manuscript.
Author Response
Dear Reviewer,
Thank you very much for your review and to address your concerns, we have added a reference on ways to extend serum persistence (lines 166-171) as well as added further examples of Nb-Fc conjugates that show great promise for therapeutical applications (lines 510-514, 583-591).
Best regards,
Bo-kyung Jin
Reviewer 2 Report
This is a high-quality review of nanobodies, which includes the history of discovery, structural features, production and applications of nanoantibodies. The manuscript is generally well-organized, and includes key and representative studies in this area. Only some minor points need revision:
1. Since the title highlights the keywords 'diagnostic and therapeutic applications', the authors may want to focus on the contents about it. Therefore, they may give more description about the examples mentioned in Chapter 5 and 6, especially the examples of therapeutic applications. In contrast, they can consider shortening Chapter 4, especially the Production and Purification part, to make the whole review article have a clear emphasis on the application of nanobodies.
2. In Chapter 7, besides the promising future, the authors may also discuss the limitations of nanobodies and nanobody-related research works, and make prospects based on that.
Author Response
Dear Reviewer,
Thank you so much for your review and to address the issues you have pointed out, we have changed the working title of the manuscript to “Nanobodies: A Review of Generation, Diagnostics and Therapeutics” as we feel that it better encapsulates our intent to deliver a review that places equal emphasis on the generation of nanobodies as well as their diagnostic/therapeutic applications. We have also added additional examples in section 6.1 and 6.3 (lines 510-514, 565-567) as well as a whole new section of 6.4 for Nb therapeutical applications against snake and scorpion venom to further highlight their therapeutical potential.
Furthermore, an extra paragraph on the limitations of using nanobodies for diagnostic and therapeutical applications has been added to the end of section 3 (lines 163-183).
Best regards,
Bo-kyung Jin
Reviewer 3 Report
This review would add to a growing list of such reviews covering the expanding variety of applications based on or containing. In this case, the authors put greater emphasis on the methods involved in the discovery, production and characterization of these valuable reagents, and then how they are applied in diagnostic and therapeutic products. As such, it does offer some useful new coverage of this exciting area of science. It is well-written and well-organized and provides an update on the latest range of applications in the pipeline. While the review effectively covers the desirable properties of nanobodies, it almost completely neglects many important limitations and weaknesses inherent to the use of nanobodies. Greater effort to cover these limitations would lead to a more balanced review.
General comments:
1. There are a variety of fundamental aspects of nanobodies that distinguish them from other binding agents and can either increase or diminish the potential suitability of these unusual antibodies for different applications. In a review that is geared towards nanobody technical and commercial applications, it seems that these features should be given greater emphasis. For example, nanobodies have a strong propensity to recognize conformational epitopes on their targets making it much more difficult to obtain high affinity nanobodies to linear peptides, lipoglycans, many non-protein venom components, denatured or unstructured proteins, misshapen proteins, etc. While this feature helps explain why nanobodies more often possess function neutralizing activities, it also reduces the target range for nanobody applications and adds to challenges when preparing conformationally native targets for immunization and screening (e.g. complex membrane receptors). Another challenge is to generate immune responses to evolutionarily conserved proteins to which camelids are largely tolerant. While nanobodies have generally proven poorly immunogenic, this remains a significant concern which still must be carefully evaluated for each new nanobody therapeutic during trials and thus perhaps should not just be dismissed as suggested in this review. Nanobodies also pose issues related to their pharmacokinetics that should be considered when developing nanobody-based therapeutics, but this is not discussed in the review.
2. The authors provide methods that, while entirely valid, seem overly prescriptive and thus they might want to cite papers that describe detailed methods from other laboratories that employ alternative methods and approaches to nanobody discovery and expression. Similarly, there is now a huge number of papers reporting discovery of bioactive nanobodies to different targets, many that might also be cited relating to cancer, autoimmunity and infectious diseases. It is understandable that the authors must down-select this list and only include citations for those papers to which they are most familiar as seems the case in this review. Nevertheless, it would perhaps be more appropriate to at least mention that the scope is much larger than might be suggested only by the selected citations.
Minor comments:
3. On line 67, author state that the ‘VH domain of a conventional mAb is incapable of functioning alone’ though VH-only sdAbs from conventional Abs have been successfully developed and tested for a long time (e.g. Ward et al. Nature, 1989, 341:544).
4. Lines 75-78 text seems contradictory as it suggests both that each CDR contributes equally to binding and that CDR3 is the main contributor to binding.
5. Line 158 states nanobodies have a ‘greater range of antigen recognition compared to their conventional mAb counterparts’ which may be true in some contexts but should also note that nanobodies have a lesser range in terms of linear or poorly structured epitopes.
6. Calpacizumab is incorrectly indicated to target TNF, and Vobarilizumab incorrectly as IL6. There may be other mislabeling so they should all be rechecked.
Author Response
Dear Reviewer,
Thank you so much for you feedback and to address your concerns, we have added an extra paragraph on the limitations of using nanobodies for diagnostic and therapeutical applications to the end of section 3 (lines 163-183).
Furthermore, we admit that the method described can be over prescriptive and thus have added a disclaimer to state that there are other protocols that exist for different purposes and cases (lines 229-231, 312-314). Section 4.2 also highlights that there are many modifications to the standard nanobody protocol that have been done such as the use of different libraries, display methods and purification methods for a variety of proteins. Lines 151-154 also draw attention to the different expression systems that can be used for nanobody expression.
Lastly, to address minor comments:
- We have removed the definitive language and that it is only typically that all 6 complementarity determining regions are required for monoclonal antibody-antigen binding.
- We have reworded that sentence to correctly express that CDR1/2/3 play an equal role for affinity in monoclonal antibodies but CDR3 is the main contributor for antigen binding affinity for nanobodies.
- We have reworded line 158 to indicate that it is only possibly in some cases that nanobodies may outperform monoclonal antibodies. We have also added the following sentence to indicate that nanobodies do not perform as well against linear epitopes when compared to monoclonal antibodies.
- We have updated the targets of Caplacizumab and Vobarilizumab to Von Willebrand factor and Il-6 receptor respectively.
Best regards,
Bo-kyung Jin